# Regulation of Embryonic Stem Cell Self-Renewal

**DOI:** 10.3390/life12081151

**Published:** 2022-07-29

**Authors:** Guofang Chen, Shasha Yin, Hongliang Zeng, Haisen Li, Xiaoping Wan

**Affiliations:** 1Shanghai Key Laboratory of Maternal Fetal Medicine, Clinical and Translational Research Center of Shanghai First Maternity and Infant Hospital, Tongji University School of Medicine, Shanghai 201204, China; sswwyql@163.com; 2Institute of Chinese Materia Medica, Hunan Academy of Chinese Medicine, Changsha 410013, China; zenghl155@163.com; 3School of Medicine, Wayne State University, Detroit, MI 48201, USA

**Keywords:** embryonic stem cells (ESCs), self-renewal, transcription factors, signaling pathways, small molecular compounds, epigenetics, non-coding RNA, cellular energetics

## Abstract

Embryonic stem cells (ESCs) are a type of cells capable of self-renewal and multi-directional differentiation. The self-renewal of ESCs is regulated by factors including signaling pathway proteins, transcription factors, epigenetic regulators, cytokines, and small molecular compounds. Similarly, non-coding RNAs, small RNAs, and microRNAs (miRNAs) also play an important role in the process. Functionally, the core transcription factors interact with helper transcription factors to activate the expression of genes that contribute to maintaining pluripotency, while suppressing the expression of differentiation-related genes. Additionally, cytokines such as leukemia suppressor factor (LIF) stimulate downstream signaling pathways and promote self-renewal of ESCs. Particularly, LIF binds to its receptor (LIFR/gp130) to trigger the downstream Jak-Stat3 signaling pathway. BMP4 activates the downstream pathway and acts in combination with Jak-Stat3 to promote pluripotency of ESCs in the absence of serum. In addition, activation of the Wnt-FDZ signaling pathway has been observed to facilitate the self-renewal of ESCs. Small molecule modulator proteins of the pathway mentioned above are widely used in in vitro culture of stem cells. Multiple epigenetic regulators are involved in the maintenance of ESCs self-renewal, making the epigenetic status of ESCs a crucial factor in this process. Similarly, non-coding RNAs and cellular energetics have been described to promote the maintenance of the ESC’s self-renewal. These factors regulate the self-renewal and differentiation of ESCs by forming signaling networks. This review focused on the role of major transcription factors, signaling pathways, small molecular compounds, epigenetic regulators, non-coding RNAs, and cellular energetics in ESC’s self-renewal.

## 1. In Vitro Culture and Stemness Maintenance of Embryonic Stem Cells

Embryonic Stem Cells (ESCs) are derived from the pluripotent inner cell mass of mammalian embryos and are capable of self-renewal and differentiation into any type of body cell [1,2,3,4]. ESCs can differentiate into cells of the ectoderm, mesoderm, and endoderm during development [5]. Multiple factors influence the developmental process. For instance, ESCs maintain their pluripotency by expressing essential transcription factors, while spatial and temporal expression and silencing of genes related to cell fate determine cell differentiation [6].

Since the establishment of the first mouse ESC line by Martin et al. in 1981, ESCs research has been at the forefront of regenerative medicine [2] after ESCs derivation from earlier stages of embryos was reported [7,8]. Unlike adult stem cells, ESCs are pluripotent and capable of differentiating into all tissues and organs of adult animals, including germ cells, under appropriate induction conditions [9]. Meanwhile, ESCs can proliferate endlessly and sustain self-renewal under the appropriate cultural conditions. Despite the rapid development in adult stem cell research in recent years, ESCs still play an irreplaceable role in gene function analysis, developmental biology, drug design, and development [10]. Therefore, research and utilization of ESCs remain one of the core issues in stem cell research.

During in vitro culture of ESCs, specific culture conditions are adopted to generate either a “steady” or “ground” state [11]. n the initial stages of establishing and maintaining ESCs, mouse fibroblasts were used as trophoblast cells and cultured on the feeder after irradiation or drug treatment. Subsequent studies revealed that the role played by trophoblast cells during this process was the production of leukemia inhibitory factor (LIF) [12,13,14]. However, in the presence of serum, LIF can replace the function of trophoblast and maintain ESCs growth without differentiation. Later, it was found that BMP4 played a beneficial role in the growth of stem cells and, as a result, replaced the serum requirements of cells in the presence of LIF [15,16]. These findings enabled researchers to grow stem cells in the presence of serum and non-trophoblast cells, or the serum replacement condition in vitro [17], thereby greatly facilitating research and application. Functional studies have confirmed that LIF activates downstream JAK/STAT3 by binding to receptor LIFR and helper receptor gp130 [18], whereas BMP4 maintains stem cell differentiation by activating Smad, which stimulates the production of helix-loop cheliform basic protein. In addition, the core transcription factors Oct3/4 [19] and Nanog [20,21] also play a key role in maintaining the ground state of stem cells. In general, several transcription factors, including Oct4, Nanog, Stat3, Sox2, c-Myc, Esrrb, Klf4, Ronin, Tcl1, Tbx3, and Rest, with or without their co-interaction factors, have been found to regulate mESC’s stemness [22,23,24].

Since the establishment of the first human ESCs (hESCs) cell line in 1998, various studies have attempted to elucidate the growth regulation of these cells, but to date, this process remains unclear. The haploid hESCs cell lines were then isolated and maintained [25]. A study reported the growth of hESCs in a serum-free culture medium containing bFGF under the conditions of trophoblast cells [26]. In addition, these cells could be successfully cultured with mouse MEF conditioned medium (MEF CM) on a cell culture plate coated with gelatin or laminin under the condition of trophoblast cells [27]. Sato et al. [28] found that activation of the Wnt signaling pathway could replace the need for culturing hESCs in MEF CM at an early stage (5–7 days) and maintain their undifferentiated state. However, it is unclear if the Wnt signaling pathway contributes to the long-term stemness of hESCs. A high level of Oct4 [29] and Nanog [28,30,31] expression in hESCs confirmed the role played by transcription factors in the preservation of stemness in these cells. Long intergenic non-coding RNAs (lincRNAs) containing more than 200 nucleotides are autonomously transcribed molecules from the intergenic regions. They have recently emerged as key modulators of the stemness of hESCs [32,33,34]. However, the mechanism to maintain stemness and self-renewal of hESCs needs further study.

## 2. Naive and Primed Pluripotent Stem Cells

In contrast to ESCs, epiblast-derived stem cells (EpiSCs) were firstly isolated from the epiblast of the post-implantation mouse embryo [35,36]. Based on cell source and functional characteristics, ESCs and EpiSCs have naïve pluripotency and primed pluripotency states, respectively [37]. There are numerous differences between primed EpiSCs and naive ESCs, ranging from the cultural conditions and regulatory network to the chromatin modifications. For example, naïve mESCs are cultured in serum/LIF or 2i/LIF media, but the maintenance of primed mEpiSCs requires the activation of Activin and FGF signaling [38]. In support of this notion, the overexpression of four transcription factors has been deployed to establish the induced pluripotent stem cells (iPSCs) [39,40].

Human pluripotent stem cells naturally appear as a native population in the blastocyst and are capable of germline and soma formation. Human naïve ESCs possess a high degree of plasticity and differentiation potential, thereby providing a novel paradigm for studying early human embryo development. In contrast to mESCs, the pluripotency of human ESCs relies on the activation of FGF/ERK signaling [36]. Notably, FGF-cultured human ESCs are ready for differentiation and thus exist in the form of primed pluripotency. So far, numerous efforts have been made to optimize the cultural conditions to facilitate the maintenance of naïve human ESCs, which produce expandable naïve extra-embryonic endoderm in response to Wnt, Nodal, and LIF signaling molecules. A culture medium containing LIF, CHIR99021, PD0325901, and other factors could be employed to maintain the naïve pluripotency of human ESCs [41,42,43,44]. Intriguingly, the molecular reprogramming trajectory of human ESCs from primed to naïve pluripotency was described [45]. However, there is still limited understanding of cellular subset changes and critical molecular events that occur during the establishment of naïve states.

This review focused on the role of major transcription factors, signaling pathways, small molecular compounds, epigenetic regulation, and non-coding RNAs in the self-renewal of ESCs.

## 3. Transcription Factors That Regulate Pluripotency in Stem Cells

Transcription factors act as molecular switches to activate or inhibit gene expression and control the fate of cells during development. Oct3/4, Sox2, and Nanog are core transcription factors involved in maintaining pluripotency of ESCs and therefore play a key role in maintaining their ground state [46]. Among these, Oct-3/4 is the most crucial factor for ESCs. It is encoded by the Pou5f1 gene and contains two POU domains. Conditional expression of Oct-3/4 in ESCs has been revealed to promote maintenance of pluripotency and multipotent differentiation. In contrast, inhibition of Oct-3/4 expression results in the loss of pluripotency and the inability to differentiate into multipotent cells. Oct4 has been reported to promote the maintenance of pluripotency in ESCs, but it cannot inhibit their differentiation. The mechanism of OCT4 alteration is similarly incapable of inhibiting ESCs differentiation [47,48]. Nanog is another important transcription factor for embryo development [49,50]. expressed in the early stages of embryonic development. It is expressed in undifferentiated mouse or human stem cells but not in their differentiated forms. Moreover, its overexpression promotes the unlimited expansion of ESCs and maintains the Oct3/4 expression level in ESCs. Sox2 and Oct-3/4 have a synergistic effect and activate Oct-Sox enhancers. This activation, in turn, enhances its expression and the expression of Nanog and Oct3/4, thus, maintaining pluripotency of ESCs [51].

These core transcription factors exist in mouse and human ESCs. They have also been demonstrated to bind to the regulatory regions of common target genes and exert synergistic effects [52,53,54]. There are at least 353 genes shared by these factors during gene regulation in hESCs. In mESCs, Oct3/4 and Sox2 also synergistically activate transcription of downstream target genes [55,56,57].

Several other transcriptional and regulatory factors are also involved in maintaining the self-renewal and pluripotency of stem cells. In the core transcriptional regulatory network, core Oct 3/4 modules include Oct4, Sox2, Nanog, BMP, LIF, Wnt signals, and downstream Smad1, STAT3, and Tcf3. These extracellular signaling pathways and core transcription factors form a regulatory network for regulating stem cell pluripotency. The transcriptional factors including Nanog, Oct3/4, STAT3, Smad1, Sox2, Zfx, c-Myc, n-Myc, Klf4, Esrrb, Tcfcp2l1, E2f1, CTCF, as well as their regulators p300 and Suz12, form various transcriptional complexes and play an important role in the regulation of the self-renewal of stem cells [58,59]. The core transcriptional regulatory clusters formed by Nanog-Oct3/4-Sox2 exhibit ESC-specific transcriptional enhancer activity, which interacts with Smad1 and STAT3 downstream of BMP and LIF signaling pathways. This process plays a key role in stabilizing transcription factor complexes, thereby enhancing the transcriptional activity of ESC-specific genes [60,61]. Wnt signaling pathways downstream of transcription factor Tcf3 act together with Oct3/4 and Nanog to occupy downstream gene promoter regions and regulate their expression. This transcription regulation of composite components is responsible for maintaining the pluripotency and a steady-state condition of ESCs [62].

Other pluripotent regulators, such as Dax1, Nac1, Zfp281, Nr5a2, and Klf4 [63,64,65], form protein complexes with Oct3/4 core regulatory modules, bind to the distal region of the transcription initiation site, and regulate Oct3/4 expression. The interaction between Oct3/4 and these transcription regulators plays a key role in the complex. Other transcription factors, c-Myc, n-Myc, E2f1, Zfx, Rex1, and Ronin, bind to sites close to the transcription initiation site and regulate the metabolism of downstream target proteins [61,66].

Notably, the expression level of core transcription factors is decisive in the homeostasis of ESCs pluripotency. For instance, the repression of Oct-3/4 expression is known to cause the loss of pluripotency and promote the dedifferentiation of ESCs into trophectoderm. In contrast, a less than twofold increase of the Oct-3/4 level results in the differentiation of ESCs to primitive endoderm or mesoderm. Continuous expression of Oct3/4 and Sox2 disrupt stem cell homeostasis and trigger stem cell differentiation [19,21]. hence supporting phase-wise activation of pluripotent genes in ESCs stemness. These reports suggest that excessive expression of core transcription factors is not a prerequisite for sustaining the stemness of stem cells. Similarly, a high expression of stemness-related genes disrupts the balance of the stem cells and most key factors (Oct3/4, Sox2, Nanog, Esrrb, Sall4, Dax1, Klf2, Klf4, Klf5, Stat3, Tet and Tcf3) [67,68,69,70,71,72,73].

### 3.1. An Important Signaling Pathway That Regulates the Fate of Embryonic Stem Cells

Signal transduction among mammalian cells can be achieved by direct contact with the adjacent cells. During this process, specific proteins secreted by the cells and a combination of chemical substances interact with cell surface receptors, activating transcription factors through downstream signaling pathways to regulate gene expression. Multiple signaling pathways regulate the fate of stem cells, including proliferation, differentiation, and maintenance of stemness. These processes can be regulated by adding small molecule modulators that activate or inhibit the signaling pathways. The important signaling pathways that regulate the fate of stem cells are described below.

### 3.2. Bidirectional Regulation of Downstream LIF-LIFR/gp130 Signaling Pathway

The activation of the Jak-Stat3 pathway is a key mediator of LIF signaling in maintaining mESCs stemness. Initial establishment and maintenance of ESCs by MEF cells required trophoblast. Later studies revealed that feeder cells maintain the pluripotency of stem cells primarily through the production of LIF, a member of IL-6 cytokine family [74]. IL-6 cytokine family includes IL-6, IL-11, LIF, Ciliary neutrophilic factor (CNTF), Cardiotrophin-1(CT-1), Oncostatin M (OSM), and other cytokines with similar biological effects. Among them, OSM, CNTF, CT-1, and LIF are known to control non-differentiation in ESCs [75,76]. As ESCs do not express an IL-6 receptor, therefore it does not exhibit the effects mentioned above.

On the other hand, IL-6 and its soluble receptor jointly act on ESCs to maintain their non-differentiation state. Different IL-6 families share a common helper receptor, gp130, a transmembrane protein with no kinase activity belonging to the cytokine receptor superfamily. It relies on the intracellular non-receptor tyrosine-protein kinase JAK for signal transmission [77]. The phosphorylation of gp130 at specific Tyr residues contributes to maintaining the undifferentiated state of mESCs by activating downstream effector molecules such as transcription factors, SH2, and STAT3 [78]. At this point, the binding of gp130 to STAT3 is mainly mediated by the Tyr phosphorylation in four YXXQ sequences, whose deletion or mutation abolishes the interaction between gp130 and STAT3. Alternatively, the dominant interfering mutant form of STAT3, known as STAT3F, is reported to abolish the self-renewal and induce the differentiation of ES cells even in the presence of LIF [79]. Consistent with this observation, the down-regulation of STAT3 expression also promotes ESCs differentiation [80,81], highlighting the indispensability of STAT3 activation for maintaining mESCs stemness.

### 3.3. LIF-LIFR/gp130 Signaling Pathway Fails to Maintain hESCs Self-Renewal

Unlike mESCs, the self-renewal of hESCs is independent of the LIF-STAT3 signaling pathway [82]. Even though human LIF could induce the phosphorylation of intracellular STAT3 via stimulating LIFRβ and gp130, functional activation of the LIF-STAT3 signaling pathway is incapable of sustaining hESCs self-renewal [83]. It has been shown that the undifferentiated state of hESCs can be sustained in feeder-free culture conditions containing particular chemical compounds [84]. BMP, TGF-β, Wnt, NODAL/ACTIVIN-A, and FGF signaling pathways have been found to play key roles in modulating hESCs stemness [85,86,87,88,89].

### 3.4. Activation of PI3K Signaling Promotes mESCs Self-Renewal

In addition to activating the JAK/Stat3 pathway, LIF can stimulate phosphoinositide 3 kinases (PI3K) in mouse and human ESCs via the gp130, representing an important mechanism for the maintenance of stemness [90]. The PI3K subsequently activates the PI (3,4,5) P3 downstream kinase (PDK1) by catalyzing the formation of PI (3,4,5) P3 from PI (4,5) P2. PDK1 phosphorylates the Akt protein upon activation, which later translocates to the nucleus to regulate downstream target genes. Consequently, LIF-induced activation of PI3K/Akt signaling promotes the self-renewal of mESCs and suppresses their differentiation. The catalytic inactivation of PI3K by gene mutations or the inhibition of PI3K activity by specific inhibitors (e.g., LY294002) leads to an increase in intracellular ERK signaling and mESCs differentiation [91]. Notably, PI3K signaling also has a great role in the modulation of ESCs proliferation. The loss of PTEN, a negative regulator of PI3K/AKT signaling, accelerates ESCs proliferation and decreases their apoptosis via enhancing Akt phosphorylation [92]. PTEN is a phosphatase that dephosphorylates PI (3,4,5) P3, and its inactivation results in a high level of intracellular PI (3,4,5) P3 responsible for the improvement of Akt activity [93,94].

### 3.5. Activation of SHP2–Ras–ERK Pathway Leads to ESCs Differentiation

In addition to activating JAK-STAT3 and PI3K pathways via gp130, LIF can activate SHP2-Ras-ERK downstream signaling pathway via gp130. The protein phosphatase SHP2 downstream of gp130 is a signal molecule containing SH2 domain, which mediates the binding between gp130 and Ras-ERK, thereby activating ERK, inhibiting self-renewal of ES cells, and accelerating their differentiation. By blocking the binding of SHP2 to gp130 through gene mutation, Ras-ERK pathway can be inhibited, and the activation time of STAT3 induced by LIF stimulation is prolonged [95]. If the junction protein between SHP2 and downstream Ras is missing, ES cells express inhibited H-Ras and block ERK activation in Ras-ERK signaling pathway [96], promoting ESCs self-renewal. Activating SHP2-Ras-ERK pathway is necessary for normal differentiation of ES cells, whereas blocking this pathway will lead to abnormal ESCs differentiation [97,98]. ESCs of mice with SHP2 mutation cannot differentiate into hematopoietic cells and fibroblasts [99].

In summary, LIF-mediated signal network maintains the pluripotency of mESCs in a bidirectional manner. When LIF acts on ES cells, it can activate the JAK-STAT3 signaling pathway through gp130 to promote cell self-renewal and activate the PI3K signaling pathway to promote cell proliferation and growth. It may also activate the SHP2-Ras-ERK pathway to promote ES cell differentiation. ES cells can only maintain self-renewal if it keeps a precise balance among the three pathways.

### 3.6. The Combination of BMP and LIF Promotes ESCs Self-Renewal in Serum-Free Condition

In the presence of serum, LIF can replace the function of trophoblast and maintain the growth of mESCs without differentiation. In the absence of serum, LIF alone cannot prevent mESCs differentiation completely, and a small number of cells will differentiate into nerve cells [15]. The above results indicate that there are other cytokines in serum that can inhibit the differentiation of ESCs into nerve cells and facilitate the self-renewal of ESCs. These cytokines play the role by combining with LIF. It was found that serum containing BMP could inhibit the differentiation of ES cells into nerve cells. BMP can inhibit the differentiation of ES cells into nerve cells. BMP 4/2 induces the expression of helix-loop cheliform (HLH) basic protein Id by activating the Smad pathway, which inhibits the differentiation of ES cells into nerve cells, suggesting that the combined action of LIF and BMP may completely block the differentiation of ES cells and maintain a high level of ES cells self-renewal [100]. As mentioned earlier, ERK phosphorylation leads to ES cell differentiation. The addition of the small molecule PD98059 inhibits the ERK upstream kinase MEK and promotes the maintenance of ESCs self-renewal [95]. BMP4 can also inhibit ESCs differentiation and promote stemness maintenance by inhibiting the phosphorylation of ERK in ESCs [101]. Figure 1 depicts LIF-mediated signaling pathways that modulate mESCs pluripotency [81,102].

### 3.7. Activation of the Wnt Signaling Pathway Boosts ESCs Self-Renewal

Activating the LIF-STAT3 signaling pathway by adding LIF can maintain mESCs self-renewal but does not affect hESCs. Activating the Wnt signaling pathway promotes self-renewal of ESC cells in humans and mice. The Wnt receptor is frizzled (Frz), a seven-fold transmembrane receptor with a cysteine-rich domain (CRD) at its extracellular N-terminus that can combine with Wnt. Frz acts on the cytoplasm of dishevel protein (Dsh or Dvl). Dsh degrades β-catenin complex (including the APC, Axin, GSK-3β, and CK1) to cut off the β-catenin degradation pathway, resulting in β-catenin accumulation in the cytoplasm, entry into the nucleus, and activation of transcription of the stemness regulating genes [103]. BIO, a specific GSK3β inhibitor of ESCs, induces activation of the β-catenin-dependent Wnt pathway and up-regulates the expression of multi-competent transcription factors, Oct-3/4, Rex-1, and Nanog, thereby promoting self-renewal of human and mouse ESCs. Activation of the Wnt signaling pathway is critical not only for maintaining the pluripotency of ESCs [26,104], but also for the maintenance of self-renewal of adult stem cells (such as skin stem cells and hematopoietic stem cells) [105,106]. Activation of the Wnt signaling pathway promotes ESCs self-renewal (Figure 2) [107].

## 4. Small Molecular Compounds That Maintain ESCs Stemness

The addition of some cytokines such as LIF and BMP4 during ESCs culture in vitro isbeneficial for the growth and stemness maintenance of ESCs. Based on an in-depth study of signal pathways related to stemness maintenance and differentiation, scientists found that adding small molecule modulators to specific signal pathway proteins benefits the stemness maintenance of ESCs. These small molecular compounds include the inhibitors CHIR99021 of GSK3β, PD0325901 of MEK, and SU5402 of FGFR signaling pathway, etc. These small molecules maintain ESCs pluripotency in the in vitro culture by inhibiting differentiation. Among them, CHIR99021 can improve the proliferation capacity of ESCs and promote the self-renewal of stem cells by inhibiting phosphorylation of downstream ERK and reducing differentiation. PD0325901 maintains the self-renewal and pluripotency of stem cells by inhibiting the MEK-ERK signaling pathway and reducing the PKB phosphorylation level. SU5402 is an FGFR inhibitor. In the FGF signaling pathway, FGF4 can activate both phosphokinase PKB and extracellular signal Ras-MRK-ERK that lead to ESCs differentiation, and the addition of compound SU5402 inhibits the above process. AG1478 is an EGFR inhibitor that maintains ESCs self-renewal by stimulating their proliferation and inhibiting MAPK signaling pathway activation [108]. Small molecule SC1 inhibits differentiation by blocking both RasGAP and ERK1 to maintain ESCs self-renewal [109]. Currently, the combination of CHIR99021 and PD0325901 at a concentration of 3 μM, known as the 2i, is commonly utilized as a key medium additive in the routine culture of ESCs. Despite this, a high concentration of these two compounds is necessary for their proper actions. The inhibitory effects of 2i on cell proliferation are a further cause for concern, as they impede research [11]. In our study group, VEGFR signaling pathway inhibitor Sunitinib was found to inhibit VEGFR to maintain the embryonic stem cell self-renewal and undifferentiated state during long-term culture in vitro [110]. With the extension of culture time and the increased volume of stem cell clones during in vitro culture of stem cells, the expression of HIF1α increased, VEGF expression was up-regulated, and cell differentiation was induced. Sunitinib inhibits the effect of VEGF by inhibiting the VEGFR signaling pathway, while feedback regulation reduces the expression of VEGF, which is beneficial for self-renewal maintenance. Identifying these small molecular compounds is conducive to further elucidating the mechanism of ESCs stemness maintenance and can be used as small molecular tools and even therapeutic leads. We have also reviewed the role of small molecular compounds in stem cell maintenance and reprogramming [111]. Therefore, uncovering additional small molecules that promote the stemness maintenance of ES cells has not only important theoretical significance but also significant application value.

## 5. Epigenetic Characteristics and Important Epigenetic Modification Factors of ES Cells

The self-renewal and differentiation of ESCs are dynamically regulated by epigenetics. Epigenetic regulatory modifications include DNA methylation modifications, histone modifications, and regulatory modifications mediated by non-coding RNAs [112]. These epigenetic regulatory modifications play an important role in regulating the self-renewal and differentiation of stem cells [113,114,115]. Generally, DNA demethylation occurs in the promoter region of important transcription factors in ESC cells, increasing transcription factor expression, while the opposite is true in differentiated cells. For example, in differentiated ESCs, the expression of core transcription factors Oct4 and Nanog is decreased after methylation of Dnmt3a and Dnmt3b [98]. Meanwhile, histone modification of ESCs is also different from that of differentiated cells. In ESCs, 87% of histones do not undergo trimethylation of H3K4 and H3K27, while differentiated cells show high trimethylation of H3K4 and H3K27 [116,117] as well as transcriptional activation of H3K4me3 and repression of H3K27me3 [118]. TRIM28 and interacting KRAB-ZNFs control the self-renewal of hESCs by modulating H3K9me3 and DNA methylation. Some epigenetic regulators modulate these processes. For example, histone regulators Jmjd1a and Jmjd2c encode histone H3K9 demineralization enzymes that prevent the accumulation of methylation in the promoter region of ESCs pluripotent genes, thus maintaining the high expression level of ES stemness genes and promoting the self-renewal of ES cells [119]. When genes Jmjd1a and Jmjd2c are knocked out or deleted, the expression of pluripotent genes, Tcl1 and Nanog will be down-regulated, and ES cells tend to differentiate. Jarid2 and Mtf2, PRC2 components, mediate histone H3K27 methylation, down-regulate Oct4 expression, and accelerate ES cell differentiation [120,121]. Meanwhile, Oct4 expression can increase the expression level of H3K9 demethylase and PRC2 complex components. The Oct4 and Sox2 downstream targeted bivalent chromatin component, Utf1, promote the ESCs pluripotency and proliferation by limiting Histone 3 lysine-27 trimethylation and PRC2 loading [122]. Currently, regulating and maintaining tumorigenicity and self-renewal in tissue stem cells as cancer stem cells are regulated by some epigenetic proteins [123,124,125,126] or non-coding RNA-mediated regulated modifications [127,128]. Our previous work uncovered that the m6 A methylation-mediated HIF-ALKBH5-SOX2 axis induces an endometrial cancer stem-like cell expansion in hypoxia [129]. Consequently, epigenetic modification factors are essential for the self-renewal of ESCs and other tissue stem cells.

## 6. The New Role of Non-Coding RNA in Stemness Maintenance

Non-coding RNAs (ncRNAs) are different types of RNA molecules that do not translate into proteins. A growing body of research shows that RNA is capable of far more than serving as a messenger or ribosomal component. Diverse non-coding RNAs are involved in gene expression, protein translation, and post-translation modification, and they play an important role in life activities [130,131,132,133,134]. Currently, many non-coding RNA studies in stemness maintenance include long non-coding RNA (lncRNA) and small RNAs (microRNA, miRNA). lncRNA is a class of RNA located between protein-coding genes and is at least 200 bases in length. miRNAs are small non-coding RNAs about 22nt long, with high interspecific conservation and spatiotemporal specificity. They play an important regulatory role in the self-renewal and differentiation of stem cells [135,136,137]. miRNAs that promote mESCs stemness maintenance include mir302 and mir290 clusters. They may shorten the G1 phase of ES cells by inhibiting the expression of key cell cycle regulators such as Cdkn1a, Rbl1, and Lats2 [138]. The core transcription factors Oct4, Sox2, and Nanog, regulate the expression of mir302 and mir290 clusters that are inhibited by the let-7 miRNAs. The let-7 miRNAs constitute a large miRNA family involving ten mature subtypes and suppress Myc expression by targeting its mRNAs [139]. Let-7 microRNAs are negative regulators of ESCs self-renewal due to their ability to inhibit the expression of Myc and other pluripotency-associated genes. For example, the up-regulation of let-7 levels in ESCs by lowering the transcription factor Lin28 has been found to restrain Myc activity and decrease the expression of downstream genes of core transcription factors, eventually resulting in the ES cell’s differentiation [140]. It has also been reported that Oct4 and Nanog regulated conserved long ncRNAs AK028326 and AK141205 modulate mESCs pluripotency [141]. In addition, the other conserved long ncRNA, TUNA, controls pluripotency and neural differentiation of ESCs [142]. Moreover, the small nucleolar RNA host gene 3 (Snhg3) [143], is essential for mESCs self-renewal and pluripotency. The stably expressed long intergenic ncRNA Cyrano maintains ESCs self-renewal by restraining mir-7 action [144]. A divergent lncRNA of lncKdm2b controls the self-renewal of ESCs by activating the expression of transcription factor Zbtb3 and ATPase activity [145]. The long ncRNAs growth arrest-specific 5 (Gas5) regulates mESCs pluripotency by the Dicer-miR291a-cMyc axis [146] and can also maintain human ESCs self-renewal via the NODAL signal [147,148]. Mir302 regulates the pluripotency of hESCs by acting on LEFTY1 and LEFTY2 regulatory factors of the Nodal signaling pathway [149]. Other studies have revealed that Oct4 can activate more than 200 nucleic acid lengths of non-coding RNA (lincRNA) and maintain the pluripotency state of ESCs [150]. Deletion of lincRNA-ROR in hESCs can lead to an increase in cell growth defects and apoptosis. The initial findings indicate that lncRNA and miRNA play an important role in the maintenance of the self-renewal capacity of ES cells. The study of the role of long non-coding RNA and microRNA in the self-renewal maintenance of ESCs is becoming an increasingly vital frontier area of stem cell research.

## 7. Cellular Energetics in Stem Cell Self-Renewal

Environmental factors and cellular metabolic processes control the metabolic homeostasis of ESCs, whose presumed primary goal is to provide essential building blocks for cell growth and functions. Accumulating evidence suggests that cellular metabolic reactions could actively regulate stem cell fate decisions (Figure 3) [151]. In addition to maintaining cellular energy, several key metabolic pathways, such as the metabolism of carbohydrates, lipids, and glutamine, have been closely associated with the fate decision of both embryonic and adult stem cell populations [151]. For instance, the mTOR pathway sensing cellular energy status has been found to enhance stem cell survival under starving conditions [152], reflecting a protective role of mTOR signaling in stem cell maintenance. As byproducts of aerobic metabolism, reactive oxygen species (ROS) are highly reduced oxygen molecules containing an unpaired electron. They have emerged as critical second messengers in the signaling cascades regulating stem cell maintenance and differentiation [153]. As the physiological regulators of ROS homeostasis, hypoxia-inducible factors (HIFs) linking glucose metabolism and redox reactions have been reported to play crucial roles in maintaining ESCs self-renewal or accelerating their vascular-lineage differentiation through a HIF1-dependent mechanism [154,155]. In addition, forkhead-box transcription factors (FoxOs) critically involved in glucose metabolism have been shown to regulate the maintenance and self-renewal of stem cells through coordinating ROS-associated cellular responses [155]. Due to their NAD+ dependency, sirtuins deacetylating specific proteins and histones function as crucial glucose and lipid metabolism regulators. They have been found to partially modulate the self-renewal of stem cells via ROS elimination [156,157]. As a key energy sensor in eukaryotic cells, AMP-activated kinase (AMPK) plays a central role in cellular energy homeostasis. Its activation triggers downstream signaling processes, ultimately regulating the homeostasis and fate of stem cells [158,159]. These observations support the notion that the cellular energetics of stem cells maintain a balance between self-renewal and differentiation.

The maintenance of pluripotency of ES cells is a multi-level regulatory process, which forms a complex regulatory network of epigenetic factors, non-coding RNAs, and various signaling pathways. Finally, it maintains a steady-state (ground state) characterized by specific stemness-related gene expression. Out of them, the core transcription factors Oct4, Sox2, Nanog and the common role of transcription factors, LIF and BMP4 regulation of signaling pathways, apparent modification, and small molecule compounds all play an important role in embryonic stem cell self-renewal. Stem cells can be cultured in vitro for a long time and maintain self-renewal, laying the groundwork for stem cell biology research and ensuring the clinical application of stem cells. Currently, scientists are conducting further research on the mechanism of self-renewal of stem cells and making new progress. These research results are bound to further promote regenerative medicine development. Studying different mechanisms that drive stem-cell fate may provide new insights into stem cell biology in disease and make new contributions to human health.

## Figures and Tables

**Figure 1 life-12-01151-f001:**
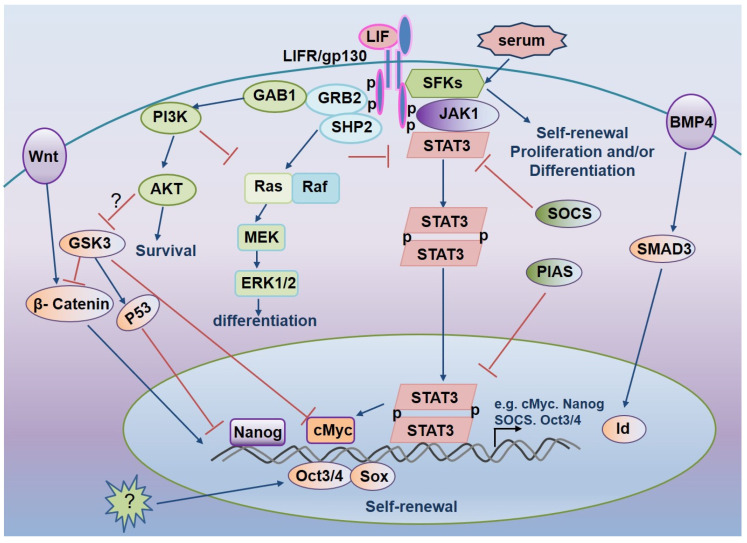
LIF-mediated signaling pathways regulate mESCs pluripotency.

**Figure 2 life-12-01151-f002:**
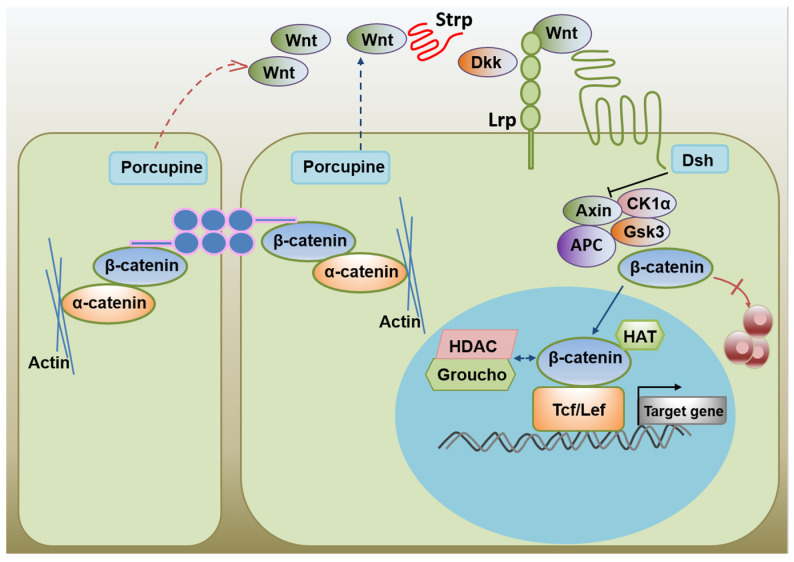
The activation of the Wnt signaling promotes ES cell self-renewal.

**Figure 3 life-12-01151-f003:**
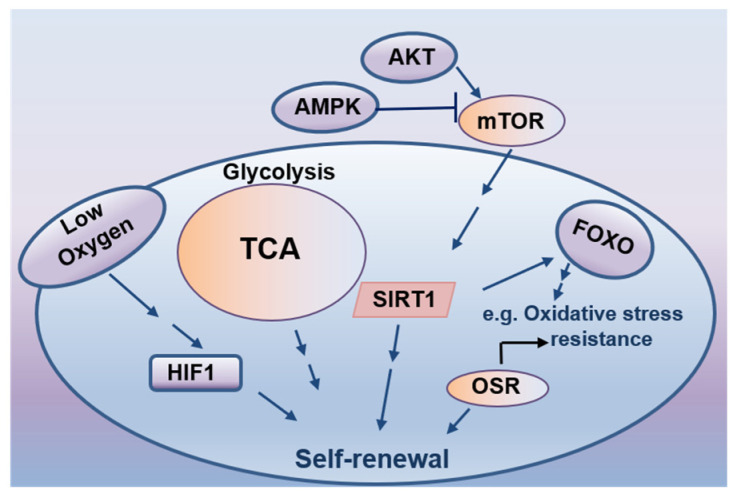
Cellular energetics in stem cell self-renewal.

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
