# Peer review of "Regulation of Embryonic Stem Cell Self-Renewal"

_life, 2022, doi:10.3390/life12081151_

Round 1
Reviewer 1 Report
The manuscript titled,” Regulation of embryonic stem cell self-renewal” is a work of importance from a developmental and basic stem cell biology perspective”.
However, the following points are suggested for the improvement of the manuscript.
1. Page-2, Para-4, “Studies have reported that excessive expression of genes involved in stemness in mESC is not a prerequisite for maintenance of pluripotency of the stem cells”. Please elaborate this section. Is there a phase-wise activation of pluripotent genes?
2. Include one paragraph of cellular energetics of PSCs involved/justifying self-renewal.
3. Include one section on naïve and primed pluripotent stem cells and how they are important for maintaining the self-renewal and pluripotency.
4. Explain in one section why pluripotency of human PSCs does not follow LIF signalling pathway.
5. Why are human ESCs called as at par with the epiblast stem cells and what such cells need to do to maintain stemness, means the signalling pathway.
6. Mention one paragraph where the stemness of human and mouse ESCs are discussed together and their respective comparative mechanisms to maintain self-renewal and stemness.
7. Include diagrams on cellular energetics and self-renewal of embryonic stem cells.
There is no link between the sections. Please ensure the same.
Author Response
The manuscript titled,” Regulation of embryonic stem cell self-renewal” is a work of importance from a developmental and basic stem cell biology perspective.
However, the following points are suggested for the improvement of the manuscript.
- Page-2, Para-4, “Studies have reported that excessive expression of genes involved in stemness in mESC is not a prerequisite for maintenance of pluripotency of the stem cells”. Please elaborate this section. Is there a phase-wise activation of pluripotent genes?
Answer: Thank you for the suggestion. We have elaborated this section. “Studies have reported that excessive expression of genes is not a prerequisite for maintenance of pluripotency involved in stemness of stem cells. The precise level of Oct-3/4 governs ES cells. Repression the expression of Oct-3/4 causes loss of pluripotency and then dedifferentiation into trophectoderm. In contrast, there is a phase-wise activation of pluripotent genes, a less than twofold increase of Oct-3/4 induces differentiation into primitive endoderm or mesoderm, thus the amount of Oct-3/4 is critical for homeostasis.”
- Include one paragraph of cellular energetics of PSCs involved/justifying self-renewal.
Answer: Thank you for the suggestion. We have added one paragraph of cellular energetics of PSCs involved in self-renewal. “Cellular energetics involved in self-renewal
Cellular energetics of stem cells including integrating environmental factors and energy metabolism pathways, which play an important role on cell-cycle regulation and self-renewal of stem cells. The cellular and systemic metabolic pathways implicated in sensing energetic status and regulating stem-cell fate in both embryonic and in adult stem cell populations [143]. The mTOR pathway is a negative regulator of autophagy, thereby plays a protective role in stem cells biology and enhances survival of the stem cells under conditions of starvation [144]. Reactive oxygen species (ROS) is the response to low oxygen levels and mediate the stem cell maintenance or lineage differentiation [145], and the hypoxia inducible factors (HIFs) play roles in maintaining the self-renewal of ESCs or accelerate vascular-lineage differentiation through HIF1-dependent mechanism [146, 147]. Forkhead-box transcription factors (FoxOs) have diverse cellular functions and are linked with ROS-regulated cellular responses to mediate stem cell self-renewal and stem cell maintenance [147]. Sirtuins which can deacetylate protein targets and histones is proposed to maintain the stem cell self-renewal through ROS elimination of hematopoietic stem cells [148, 149] and in the NSCs maintenance [150, 151]. AMP-activated kinase (AMPK) is the key energy sensor in eukaryotic cells, AMPK activation leads to activation of downstream processes and stem cell homeostasis. The tumor suppressor gene LKB1 can activate AMPK pathway and related kinases to control of the stem cell state [152, 153]. These cellular energetics of stem cells keep balance between self-renewal and differentiation.”
- Include one section on naïve and primed pluripotent stem cells and how they are important for maintaining the self-renewal and pluripotency.
Answer: Thank you for the suggestion. We have added one section on naïve and primed pluripotent stem cells and how they are important for maintaining the self-renewal and pluripotency.
“Naive and primed pluripotent stem cells maintaining pluripotency
The first epiblast-derived stem cells (EpiSCs), which distinguish from ESCs, are new type of stem cells isolated from the post-implantation epiblast in mice in 2007 [35, 36]. Then many groups were building the isolation of induced pluripotent stem cells (iPSCs) with four transcription factors [37, 38] that forced the multiple stable pluripotent states, leading to the proposal of naive and primed pluripotent states, representing the cellular state of the preimplantation blastocyst inner cell mass, and the post-implantation epiblast cells [39]. Different culture conditions and functional capacities between naive and primed pluripotency from to gene expression profiles and chromatin modification states. Naive mESCs cultured in serum/LIF or 2i/LIF medium, Primed mEpiSCs are require Activin and FGF signaling [40].
Human naive pluripotent stem cells have higher plasticity and differentiation potential which providing a unique model for early embryo development research and application. Human ESCs emerge as a naive founder population in the blastocyst, which have germline and soma formation capacity, and then undergo lineage differentiation. Human ESCs and epiblast-derived stem cells (EpiSCs) represent the initial naive and final primed phases of pluripotency. In human, ESCs maintain pluripotency by FGF/ERK activation [36], cells cultured in the presence of FGF are ‘primed’ for differentiation. Other reports have identified culture conditions include 2i/LIF and additional factors for a more naïve phase of human pluripotency, EpiSCs Naive human ESCs respond to the Wnt, Nodal and LIF signaling to produce expandable naive extra-embryonic endoderm [41-44]. A recent study is the first to depict the molecular reprogramming trajectory of human cells from primed pluripotent to naive pluripotent [45]. However, there is limited understanding of the key molecular events that occur during the establishment of naive states. Changes in cellular subsets and key molecular events during the induction of primed to naive pluripotent remain unclear.”
- Explain in one section why pluripotency of human PSCs does not follow LIF signaling pathway.
Answer: Thank you for the suggestion. We have added one section to explain human PSCs does not follow LIF signaling pathway. “LIF-LIFR /gp130 signaling pathway fails to maintain human ESCs self-renewal. Unlike mESCs, hESCs self-renewal is not regulated by LIF-STAT3 signaling pathway, LIF-STAT3 signal maintains mESC self-renew instead of hESC [73]. The feeder-free culture conditions maintain hESCs in an undifferentiated state, and don’t follow LIF-LIFR /gp130 signaling pathway [74]. Human ESCs express LIFRβ and gp130, and the human LIF can also induce STAT3 phosphorylation, then translocate into nuclear, but the functional activation of the LIF-STAT3 signaling pathway fails to maintain hESCs self-renewal [75]. Signaling pathways include BMP, TGF-β, Wnt, and NODAL/ACTIVIN-A-activated FGF signaling pathway appear to be biologically relevant for stem cell functions and their receptors in hESCs have potential effects for their undifferentiated state [76-80].”
- Why are human ESCs called as at par with the epiblast stem cells and what such cells need to do to maintain stemness, means the signaling pathway.
Answer: Thank you for the question. Human naive pluripotent stem cells have higher plasticity and differentiation potential which providing a unique model for early embryo development research and application. Human ESCs emerge as a naive founder population in the blastocyst, which have germline and soma formation capacity, and then undergo lineage differentiation. Human ESCs and epiblast-derived stem cells (EpiSCs) represent the initial naive and final primed phases of pluripotency. In human, ESCs maintain pluripotency by FGF/ERK activation [36], cells cultured in the presence of FGF are ‘primed’ for differentiation. Other reports have identified culture conditions include 2i/LIF and additional factors for a more naïve phase of human pluripotency, EpiSCs Naive human ESCs respond to Wnt, Nodal and LIF signaling to produce expandable naive extra-embryonic endoderm [41-44]. A recent study is the first to depict the molecular reprogramming trajectory of human cells from primed pluripotent to naive pluripotent [45]. However, there is limited understanding of the key molecular events that occur during the establishment of naive states. Changes in cellular subsets and key molecular events during the induction of primed to naive pluripotent remain unclear.
- Mention one paragraph where the stemness of human and mouse ESCs are discussed together and their respective comparative mechanisms to maintain self-renewal and stemness.
Answer: Thank you for the suggestion.
“The mESCs maintain ESC growth in the presence of LIF downstream JAK/STAT3/gp130 pathway, transcription factors and other factors in the self-renewal of embryonic stem cells. While hESC growth in a serum-free culture medium containing bFGF and the Wnt signaling pathway maintain their undifferentiated state.”
- Include diagrams on cellular energetics and self-renewal of embryonic stem cells.
Answer: Thank you for the suggestion. We have added diagrams on cellular energetics and self-renewal of embryonic stem cells.
Figure 3. Cellular energetics involved in self-renewal

Reviewer 2 Report
Reviewer Comments
Journal: Life
Paper number: Life-1780001
Paper title: Regulation of embryonic stem cell self-renewal
Authors: Guofang Chen, Shasha Yin and Hong liang Zeng
General Comment:
This review focused on the role of major transcription factors, signaling pathways, small molecular compounds, epigenetic regulation and non-coding RNA in ESC self-renewal. There are numerous strengths to this study, including in vitro and in vivo data and well-updated references. However, it requires some additional revision.
Minor issues
My minor criticisms relate to the following issues:
- Could you rephrase the following sentence, page 1 of 14 for example: Embryonic stem cells (ESCs), a group of cells derived from pluripotent inner cell mass of blastocyst, are the type of cell capable of self-renewal and multi-directional differentiation [1, 1A, B, C].
- Could you the following references:
1A. Evans MJ, Kaufman MH. Establishment in culture of pluripotential cells from mouse embryos. Nature 1981; 292:154–156 has 11483 citations
1B. Thomson JA, Itskovitz-Eldor J, Shapiro SS, et al. Embryonic stem cell lines derived from human blastocysts. Science 1998; 282: 1145–1147. has 19689 citations
1C. Guanyi Huang, Shoudong Ye, Xingliang Zhou, Dahai Liu, and Qi-Long Ying. Molecular basis of embryonic stem cell self-renewal: from signaling pathways to pluripotency network. Cell Mol Life Sci. 2015 May; 72(9): 1741–1757.
- The title should be the same and in bold and font as the others, page 2 of 14: Transcription factors that regulate pluripotency in stem cells
- This sentence should not be in bold and a dot should be added at the end of the sentence, page 4 of 14. LIF-mediated activation of the Jak-Stat3 pathway promotes mESC stemness maintenance.
- A dot should be added at the end of the sentence, page 4 of 14: Phosphorylation of gp130 at specific Tyr residues sites activates downstream signaling molecules such as transcription factors, SH2, STAT3 to initiate expression of target genes [69].
- Could you rephrase the following sentence, page 4 of 14: Deletions or mutations in these sites hinders the gp130 binding and STAT3 activation [70], and the self-renewal of the resultant ESCs cannot be maintained.
- Could you rephrase the following sentence, page 4 of 14: LIF-STAT3 signal maintains mESC self-renew and cannot support self-renewal of hESC [16, 24].
- Could you rephrase the following sentence, page 4 of 14: PI3K catalyzes phosphorylation of PI (4,5) P2 to form PI (3,4,5) P3, while Akt is phosphorylated by PDK1 downstream of PI (3,4,5) P3 to enable downstream targets for regulation of various life activities.
- Could you remove the comma after PTEN, page 4 of 14: PTEN is a phosphatase that can dephosphorylate PI (3,4,5) P3 thus inhibiting the PI3K signaling pathway [75, 76].
- Could you rephrase the following sentence, page 4 of 14: Knocking out of PTEN, which negatively regulates PI3K/AKT signalling by dephosphorylating PtdIns(3,4,5)P3, potentially could induce high levels of intracellular PI (3,4,5) P3 and as well as phosphorylation of downstream signaling molecule Akt leading to accelerated cell proliferation and reduced cell apoptosis, maintaining the pluripotency of ESC [75, 77].
- Authors should head towards the paragraph, page 4 of 14: In addition to activating the JAK-STAT3 pathway and PI3K pathway through gp130, LIF can also activate another major downstream signaling pathway, SHP2-Ras-ERK, through gp130.
- Authors should head towards the paragraph, page 5 of 14: BMP combined with LIF promotes self-renewal of ES cells in serum free condition
- Authors should remove that from the paragraph, page 7 of 14: Small molecular compounds promote stemness maintenance.
- Authors should add a reference, page 7 of 14: Currently, 2i (a combination of two small molecule inhibitors CHIR99021 3M and PD0325901 3M) has been used as an important medium additive in stem cell culture. However, the combination of these two compounds has a large concentration and an inhibitory effect on cell proliferation, which brings some inconvenience to the research [].
Author Response
- Could you rephrase the following sentence, page 1 of 14 for example: Embryonic stem cells (ESCs), a group of cells derived from pluripotent inner cell mass of blastocyst, are the type of cell capable of self-renewal and multi-directional differentiation [1, 1A, B, C].
Answer: Thank you for the suggestion. We have rephrased the sentence with “Embryonic stem cells (ESCs), a group of cells derived from pluripotent inner cell mass of the blastocyst, are the type of cells capable of self-renewal and multi-directional differentiation [1-4].”
- Could you the following references:
1A. Evans MJ, Kaufman MH. Establishment in culture of pluripotential cells from mouse embryos. Nature 1981; 292:154–156 has 11483 citations
1B. Thomson JA, Itskovitz-Eldor J, Shapiro SS, et al. Embryonic stem cell lines derived from human blastocysts. Science 1998; 282: 1145–1147. has 19689 citations
1C. Guanyi Huang, Shoudong Ye, Xingliang Zhou, Dahai Liu, and Qi-Long Ying. Molecular basis of embryonic stem cell self-renewal: from signaling pathways to pluripotency network. Cell Mol Life Sci. 2015 May; 72(9): 1741–1757.
Answer: Thank you for the suggestion. We have rephrased the sentence and references, page 1 of 14 for example: Embryonic stem cells (ESCs), a group of cells derived from pluripotent inner cell mass of blastocyst, are the type of cell capable of self-renewal and multi-directional differentiation [1-4]
- Martin, G.R., Isolation of a pluripotent cell line from early mouse embryos cultured in medium conditioned by teratocarcinoma stem cells. Proc Natl Acad Sci U S A, 1981. 78(12): p. 7634-8.
- Evans, M.J. and M.H. Kaufman, Establishment in culture of pluripotential cells from mouse embryos. Nature, 1981. 292(5819): p. 154-6.
- Thomson, J.A., et al., Embryonic stem cell lines derived from human blastocysts. Science, 1998. 282(5391): p. 1145-7.
- Huang, G., et al., Molecular basis of embryonic stem cell self-renewal: from signaling pathways to pluripotency network. Cell Mol Life Sci, 2015. 72(9): p. 1741-57.
- The title should be the same and in bold and font as the others, page 2 of 14: Transcription factors that regulate pluripotency in stem cells
Answer: Thank you for the suggestion. We have changed the bold of the title.
- This sentence should not be in bold and a dot should be added at the end of the sentence, page 4 of 14. LIF-mediated activation of the Jak-Stat3 pathway promotes mESC stemness maintenance.
Answer: Thank you for the suggestion. We have changed the bold and added a dot.
- A dot should be added at the end of the sentence, page 4 of 14: Phosphorylation of gp130 at specific Tyr residues sites activates downstream signaling molecules such as transcription factors, SH2, STAT3 to initiate expression of target genes [69].
Answer: Thank you for the suggestion. We have changed the bold and added a dot.
- Could you rephrase the following sentence, page 4 of 14: Deletions or mutations in these sites hinders the gp130 binding and STAT3 activation [70], and the self-renewal of the resultant ESCs cannot be maintained.
Answer: Thank you for the suggestion. We have rephrased the sentence with “Deletions or mutations in these sites hinder the gp130 binding to STAT3 and downstream activation, the dominant interfering mutant of STAT3, STAT3F, abrogated self-renewal and promoted differentiation of ES cells growing in the presence of LIF”.
- Could you rephrase the following sentence, page 4 of 14: LIF-STAT3 signal maintains mESC self-renew and cannot support self-renewal of hESC [16, 24].
Answer: Thank you for the suggestion. We have rephrased the sentence with the new paragraph. “LIF-LIFR /gp130 signaling pathway fails to maintain human ESCs self-renewal. Unlike mESCs, hESCs self-renewal is not regulated by LIF-STAT3 signaling pathway, LIF-STAT3 signal maintains mESC self-renew instead of hESC [73]. The feeder-free culture conditions maintain hESCs in an undifferentiated state, and don’t follow LIF-LIFR /gp130 signaling pathway [74]. Human ESCs express LIFRβ and gp130, and the human LIF can also induce STAT3 phosphorylation, then translocate into nuclear, but the functional activation of the LIF-STAT3 signaling pathway fails to maintain hESCs self-renewal [75]. Signaling pathways include BMP, TGF-β, Wnt, and NODAL/ACTIVIN-A-activated FGF signaling pathway appear to be biologically relevant for stem cell functions and their receptors in hESCs have potential effects for their undifferentiated state [76-80].”
- Could you rephrase the following sentence, page 4 of 14: PI3K catalyzes phosphorylation of PI (4,5) P2 to form PI (3,4,5) P3, while Akt is phosphorylated by PDK1 downstream of PI (3,4,5) P3 to enable downstream targets for regulation of various life activities.
Answer: Thank you for the suggestion. We have rephrased the sentence. “PI3K catalyzes phosphorylation of PI (4,5) P2 to form PI (3,4,5) P3, and activates the PI (3,4,5) P3 downstream kinase PDK1, PDK1 phosphorylates Akt, then activates the PI3K/ Akt pathway and acts on downstream targets for regulation of various life activities.”
- Could you remove the comma after PTEN, page 4 of 14: PTEN is a phosphatase that can dephosphorylate PI (3,4,5) P3 thus inhibiting the PI3K signaling pathway [75, 76].
Answer: Thank you for the suggestion. We have removed the comma after PTEN.
- Could you rephrase the following sentence, page 4 of 14: Knocking out of PTEN, which negatively regulates PI3K/AKT signaling by dephosphorylating PtdIns (3,4,5) P3, potentially could induce high levels of intracellular PI (3,4,5) P3 and as well as phosphorylation of downstream signaling molecule Akt leading to accelerated cell proliferation and reduced cell apoptosis, maintaining the pluripotency of ESC [75, 77].
Answer: Thank you for the suggestion. We have rephrased the sentence. “Knocking out of PTEN, which negatively regulates PI3K/AKT signaling by dephosphorylating PI (3,4,5) P3, potentially could induce high levels of intracellular PI (3,4,5) P3 and as well as phosphorylation of downstream signaling molecule Akt leading to accelerated cell proliferation and reduced cell apoptosis, maintaining the pluripotency of ESC.”
- Authors should head towards the paragraph, page 4 of 14: In addition to activating the JAK-STAT3 pathway and PI3K pathway through gp130, LIF can also activate another major downstream signaling pathway, SHP2-Ras-ERK, through gp130.
Answer: Thank you for the suggestion. We have headed towards the paragraph.
- Authors should head towards the paragraph, page 5 of 14: BMP combined with LIF promotes self-renewal of ES cells in serum free condition
Answer: Thank you for the suggestion. We have headed towards the paragraph.
- Authors should remove that from the paragraph, page 7 of 14: Small molecular compounds promote stemness maintenance.
Answer: Thank you for the suggestion. We have removed the paragraph.
- Authors should add a reference, page 7 of 14: Currently, 2i (a combination of two small molecule inhibitors CHIR99021 3M and PD0325901 3M) has been used as an important medium additive in stem cell culture. However, the combination of these two compounds has a large concentration and an inhibitory effect on cell proliferation, which brings some inconvenience to the research [].
Answer: Thank you for the suggestion. We have added a reference. “2i (a combination of two small molecule inhibitors CHIR99021 3μM and PD0325901 3μM) has been used as an important medium additive in stem cell culture [11]. However, the combination of these two compounds has a large concentration and an inhibitory effect on cell proliferation, which brings some inconvenience to the research [102]”
Reviewer 3 Report
In the review titled: "Regulation of embryonic stem cell self-renewal" the authors give an overview on the effects of transcription factors, signaling pathways, small molecules, and ncRNA on stem cell renewal. They cover in detail well established mechanisms, and bring to light the involvement of ncRNA. The availability of such a review will be beneficial not only for the stem cell community, but also for the scientific community at large.
While they cover most of the finding in the field, few things have been left out. Specifically the role of TGFb signaling, and ROCK-inhibitors during ESC proliferation and stem cell renewal.
Minor revisions:
1. Explain the correlation between let-7 (miRNA) and Myc.
2. Describe lincRNA. In the second to last paragraph, the authors mention lincRNAs, but fail to give a brief description of what they are, and how they differ from other ncRNAs.
Author Response
In the review titled: "Regulation of embryonic stem cell self-renewal" the authors give an overview on the effects of transcription factors, signaling pathways, small molecules, and ncRNA on stem cell renewal. They cover in detail well established mechanisms, and bring to light the involvement of ncRNA. The availability of such a review will be beneficial not only for the stem cell community, but also for the scientific community at large.
While they cover most of the finding in the field, few things have been left out. Specifically the role of TGFb signaling, and ROCK-inhibitors during ESC proliferation and stem cell renewal.
Minor revisions:
- Explain the correlation between let-7 (miRNA) and Myc.
Answer: Thank you for the suggestion. We have explained the correlation between let-7 (miRNA) and Myc. “let-7 is a large family of miRNAs, which silences self-renewal by suppressing many of the downstream targets that are indirectly activated in ESCs and promotes de-differentiation. Myc was identified as a let-7 target and significantly downregulated by let-7c [132]. Transcription factor Lin28 inhibits the expression of let-7, and down-regulation of Lin28 can increase the level of let-7 in ES cells, thereby inhibiting the activity of Myc, reducing the expression of downstream genes of core transcription factor, and accelerating the differentiation of ES cells [133]”
- Describe lincRNA. In the second to last paragraph, the authors mention lincRNAs, but fail to give a brief description of what they are, and how they differ from other ncRNAs.
Answer: Thank you for the suggestion. We have described the lincRNA. “lincRNAs is one type of the non-coding RNAs (ncRNAs), which derived from thousands of genomes loci and enriched in transposable elements (TEs), played key roles in cellular process and gene regulation.”
Reviewer 4 Report
Dear Sir/Madam
The results presented in the manuscript entitled “Regulation of embryonic stem cell self-renewal” are in a logical sequence to that contain data to inform the readers. The manuscript is interesting to publication in “Life” after minor revision.
1) Some grammatical points can be seen in the text of the manuscript.
Kind Regards
Author Response
The results presented in the manuscript entitled “Regulation of embryonic stem cell self-renewal” are in a logical sequence to that contain data to inform the readers. The manuscript is interesting to publication in “Life” after minor revision.
1) Some grammatical points can be seen in the text of the manuscript.
Answer: Thank you for the suggestion. We have revised the manuscript, and conducted additional scientific editing of both language as well as grammatical points in this revised version to make it more fluent.